# Therapeutic adenine base editing of human hematopoietic stem cells

Jiaoyang Liao [1,5], Shuanghong Chen [1,5,6] ✉, Shenlin Hsiao[1,5], Yanhong Jiang[1,5], Yang Yang[2], Yuanjin Zhang[3], Xin Wang[3], Yongrong Lai[2], Daniel E. Bauer[4] & Yuxuan Wu [1,6] ✉

In β-thalassemia, either γ-globin induction to form fetal hemoglobin (α2γ2) or β-globin repair to restore adult hemoglobin (α2β2) could be therapeutic. ABE8e, a recently evolved adenine base editor variant, can achieve efficient adenine conversion, yet its application in patient-derived hematopoietic stem cells needs further exploration. Here, we purified ABE8e for ribonucleoprotein electroporation of β-thalassemia patient CD34[+] hematopoietic stem and progenitor cells to introduce nucleotide substitutions that upregulate γ-globin expression in the *BCL11A* enhancer or in the *HBG* promoter. We observed highly efficient on-target adenine base edits at these two regulatory regions, resulting in robust γ-globin induction. Moreover, we developed ABE8e-SpRY, a near-PAMless ABE variant, and successfully applied ABE8e-SpRY RNP to directly correct HbE and IVS II-654 mutations in patient-derived CD34[+] HSPCs. Finally, durable therapeutic editing was produced in self-renewing repopulating human HSCs as assayed in primary and secondary recipients. Together, these results support the potential of ABE-mediated base editing in HSCs to treat inherited monogenic blood disorders.

In β-thalassemia, insufficient production of the β-globin molecule results in an excess of unpaired α-globin chains, impairing the maturation and leading to death of erythroid precursors. Both β-globin repair and γ-globin induction can reduce the globin chain imbalance. Induction of erythrocyte fetal hemoglobin (HbF) comprising α- and γ-globins (α2γ2) is a universal therapeutic strategy for ameliorating severe and potentially life-threatening manifestations of transfusion-dependent β-thalassemia (TDT) and sickle cell disease (SCD)[1–5]. The switch from fetal to adult hemoglobin (HbA, α2β2) around birth relies on repression or silencing of the paralogous γ-globin genes (*HBG1/2*)[6]. Dense clustering of the point mutations in *HBG1/2* promoters associated with HPFH (hereditary persistence of fetal hemoglobin) were discovered to be binding sites for transcriptional repressors that result

in autonomous silencing of the γ-globin gene[1,7,8]. BCL11A and ZBTB7A (also known as LRF) are two identified major γ-globin gene repressors and account for the majority of γ-globin silencing[9–12]. And genome-wide association studies, lentiviral-pooled sgRNA screening assay and recent preliminary clinical reports demonstrated that the erythroid-specific enhancer region of *BCL11A* was required for its expression[6,13,14], thus disruption of this region could restore γ-globin synthesis and HbF production[3–5]. Therefore, there are two major strategies for elevating HbF levels in adulthood, down-regulating expression of transcriptional repressors such as BCL11A[13,14] or mimicking the naturally occurring HPFH-associated mutations in *HBG* promoters to disrupt binding of transcriptional repressors[7,8]. Previously, we and others have determined that both CRISPR-Cas9-mediated biallelic indels and CBE-

[1]Shanghai Frontiers Science Center of Genome Editing and Cell Therapy, Shanghai Key Laboratory of Regulatory Biology, Institute of Biomedical Sciences and School of Life Sciences, East China Normal University, Shanghai, China. [2]Department of Hematology, The First Affiliated Hospital of Guangxi Medical University, Nanning, Guangxi, China. [3]Shanghai Key Laboratory of Regulatory Biology, Institute of Biomedical Sciences and School of Life Sciences, East China Normal University, Shanghai, China. [4]Cancer and Blood Disorders Center, Dana-Farber Cancer Institute and Boston Children's Hospital, Harvard Medical School, Boston, MA, USA. [5]These authors contributed equally: Jiaoyang Liao, Shuanghong Chen, Shenlin Hsiao, Yanhong Jiang. [6]These authors jointly supervised this work: Shuanghong Chen, Yuxuan Wu. ✉e-mail: shchen@bio.ecnu.edu.cn; yxwu@bio.ecnu.edu.cn

induced base editing either within a GATA1 binding motif at the +58 *BCL11A* erythroid enhancer or in the *HBG1/2* promoters of human LT-HSCs could lead to potent therapeutic HbF level for the treatment of β-hemoglobinopathies[3–5,8,15–17]. Adenine base editor (ABE) can introduce A•T to G•C conversion efficiently and cleanly, with less off-target transcriptome modification than CBE and with minimal undesired indels compared to current Cas9 nuclease-based method[18–21]. Moreover, recent further evolution created new generation of adenine base editor (ABE8e) with enhanced activity and Cas domain compatibility[22,23]. However, the feasibility of ABE8e-mediated base editing in HSCs to enable durable modification of blood cells remains uncertain.

In this study, we perform electroporation of purified ABE8e protein with chemically modified synthetic sgRNAs (MS-sgRNAs) as ribonucleoprotein (RNP) complexes targeting +58 *BCL11A* enhancer and *HBG1/2* promoters, respectively or combined, in human CD34+ hematopoietic stem and progenitor cells (HSPCs) (Fig. 1 and Supplementary Fig. 1), observing both single and multiplex editing lead to robust A > G base edits. We confirm that ABE8e-sgRNA RNP complex can achieve efficient multiplex editing in both healthy and β-thalassemia human HSPCs, efficiently inducing γ-globin. Both primary and secondary transplants have comparable editing efficiencies to input, indicating that ABE8e can efficiently edit long-term hematopoietic stem cells. After strict detection by deep sequencing, no harmful off-target editing is found. Comparing mRNA and RNP delivery methods, we find that RNP delivery reduces gRNA-dependent DNA off-targeting without compromising editing efficiency. Notably, we also successfully repair HbE and IVS II-654 causative mutations in situ using a near PAM-less variant ABE8e-SpRY, providing support for the use of ABE to treat genetic diseases previously inaccessible.

## Results

### ABE8e produces efficient A to G editing resulting in γ-globin induction

We synthesized five MS-sgRNAs targeting the +58 *BCL11A* enhancer including four sgRNAs with editable adenines within a core TGN$_{7–9}$WGATAR half E-box/GATA1 binding motif (Fig. 1a). After electroporation of ABE8e with each sgRNA as RNP complex into normal human peripheral-blood-mobilized CD34+ HSPCs, we observed various editing efficiencies (Fig. 1b). The sg1619 (A$_5$: mean 83.2%, A$_8$: mean 67.4%, A$_9$: mean 31.0%) and sg1620 (A4: mean 75.7%, A7: mean 91.7%, A9: mean 21.0%) were more efficient than sg1617 (A$_3$: mean 2.4%, A$_4$: mean 7.6%, A$_6$: mean 37.7%), sg1618 (A$_9$: mean 27.8%, A$_{12}$: mean 26.1%, A$_{14}$: mean 12.2%) and sg1621 (A$_2$: mean 32.0%, A$_4$: mean 83.6%, A$_{11}$: mean 23.6%) (Fig. 1b). In addition, the edited adenines in the sg1617 targeting sequence are outside the core half E-box/GATA motif at the +58 *BCL11A* enhancer, which may be not sufficient for robust HbF induction (Fig. 1a). After erythroid differentiation, we found that the editing efficiency in the GATA motif was positively correlated with the level of γ-globin induction (Fig. 1b, c). Notably, the γ-globin mRNA level in sg1620-edited and sg1619-edited cells were nearly four-fold higher than unedited controls (Fig. 1c). sg1617-edited cells showed comparable γ-globin expression to unedited controls, suggesting base edits outside the WGATAR motif were of limited functional significance. These results suggest that ABE8e is able to efficiently edit the core half E-box/GATA motif at the +58 *BCL11A* enhancer resulting in potent γ-globin induction in HSPCs.

Given previous report that electroporation of ABE mRNA with sgRNA is more effective than delivery of protein-RNA (RNP) complexes for editing human CD34+ HSPCs while the guide RNA-dependent DNA off-target editing is greatly reduced when base editor with sgRNA is delivered as RNP[24], we examined both on-target and off-target base editing by either the ABE8e + sg1620 RNP or ABE8e mRNA with sg1620. The RNP offered comparable on-target editing efficiencies to mRNA delivery, despite the more transient lifetime of RNP than mRNA (Fig. 1d

and Supplementary Fig. 2). As anticipated, the guide RNA-dependent DNA off-target editing by the RNP was lower than that resulted from the mRNA, probably because of the shorter duration of exposure (Fig. 1d and Supplementary Fig. 2). Quantitative evaluation of RNA single-nucleotide variations (SNVs) by whole-transcriptome RNA sequencing (RNA-seq) found that ABE8e RNP and mRNA delivery methods generated similar levels of guide RNA-independent off-target RNA SNVs (Fig. 1e and Supplementary Fig. 3). The above observations reveal that for applications in which DNA off-target editing must be minimized, we recommend the use of ABE8e RNP delivery when possible.

The clinical severity of β-hemoglobinopathies is alleviated by high γ-globin gene expression associated with HPFH, caused by *HBB* cluster deletions or point mutations in the *HBG1* or *HBG2* promoters (from −210 to −100 nucleotides upstream the *HBG* transcription start sites)[8]. Strikingly, naturally occurring HPFH mutations within the −115 region (−117, −114, and the 13 bp deletion) that span the BCL11A-binding site (TGACCA: from −118 to −113) impair BCL11A's ability to directly bind to the promoter and thus elevate HbF levels[8,25,26], prompting gene editing approaches to disrupt this motif for treating β hemoglobinopathies. To assess whether ABE8e has the ability to efficiently convert targeted A•T base pairs to G•C at the BCL11A-binding sites in the *HBG* promoters, we designed two sgRNAs (sgHBGsense, sgHBGsite1) and electroporated each RNP complex to edit HSPCs from healthy donors (Fig. 1f). ABE8e-sgHBGsense mediated efficient base editing at protospacer position 5, 8, 9 and 11 averaging 74.1%, 79.2%, 18.2%, and 52.7%, respectively, and reactivated γ-globin expression by 3.02-fold compared to mock (Fig. 1g, h). The sgHBGsite1 also offered high editing efficiency (A$_7$: mean 81.0%, A$_8$: mean 21.4%), whereas induced much lower γ-globin mRNA level than sgHBGsense, probably due to its edited adenines at target loci were not as sufficient as that of sgHBGsense to disrupt the binding site of BCL11A (Fig. 1f–h).

Using the Cas-OFFinder tool, 28 potential genomic off-target sites with three or fewer mismatches to the sg1620-target sequence and 10 potential genomic off-target sites with three or fewer mismatches relative to the sgHBGsense-target site were identified. At three sg1620-dependent candidate off-target sites nominated by Cas-OFFinder, hg38 chr3-197489118-197489140 (BCL11A_OFT6), hg38 chr4-4338932-4338954 (BCL11A_OFT9) and hg38 chr4-35881812-35881834 (BCL11A_OFT11), we detected a difference between control and edited samples, with RNP editing efficiencies of 21.4%, 40.2% and 8.1%, respectively (Supplementary Fig. 2a). None of these sites overlapped coding, regulatory or conserved sequence elements. For sgHBGsense-dependent off-target base editing, we found that 1 off-target base edit (hg38 chr8-145029260-145029282, HBG_OFT2) which was present in edited samples and unedited samples at similar frequency suggesting this may be naturally occurring A•T-to-G•C single-nucleotide polymorphisms (SNPs) (Supplementary Fig. 2b).

Because the *BCL11A* enhancer and *HBG1/2* promoter regions are distinct targets for inducing γ-globin expression, and single therapeutic base editing of either *BCL11A* enhancer or *HBG1/2* promoters resulted in effective γ-globin de-repression, we hypothesized that multiplex base editing with ABE8e targeting both the *BCL11A* enhancer and the *HBG1/2* promoters could produce additive effects. We performed multiplex editing and observed similar base edits at each target site as compared to single editing (Fig. 1i). Intriguingly, multiplex editing resulted in an increased proportion of γ-globin mRNA (by 1.52-fold and 1.88-fold, respectively, Fig. 1j) and led to elevated γ-globin protein level (by 1.70-fold and 1.85-fold, respectively, Fig. 1k and Supplementary Fig. 4) compared with either sg1620 or sgHBGsense editing alone. Next, we electroporated human CD34+ HSPCs with ABE8e + sg1620 RNP and ABE8e + sgHBGsense RNP assembled in equimolar concentrations ranging from 3.75 to 45 μM (Supplementary Fig. 5a). We found base editing was dose dependent, with 30–45 μM each RNP producing highest base edits and there was a strong

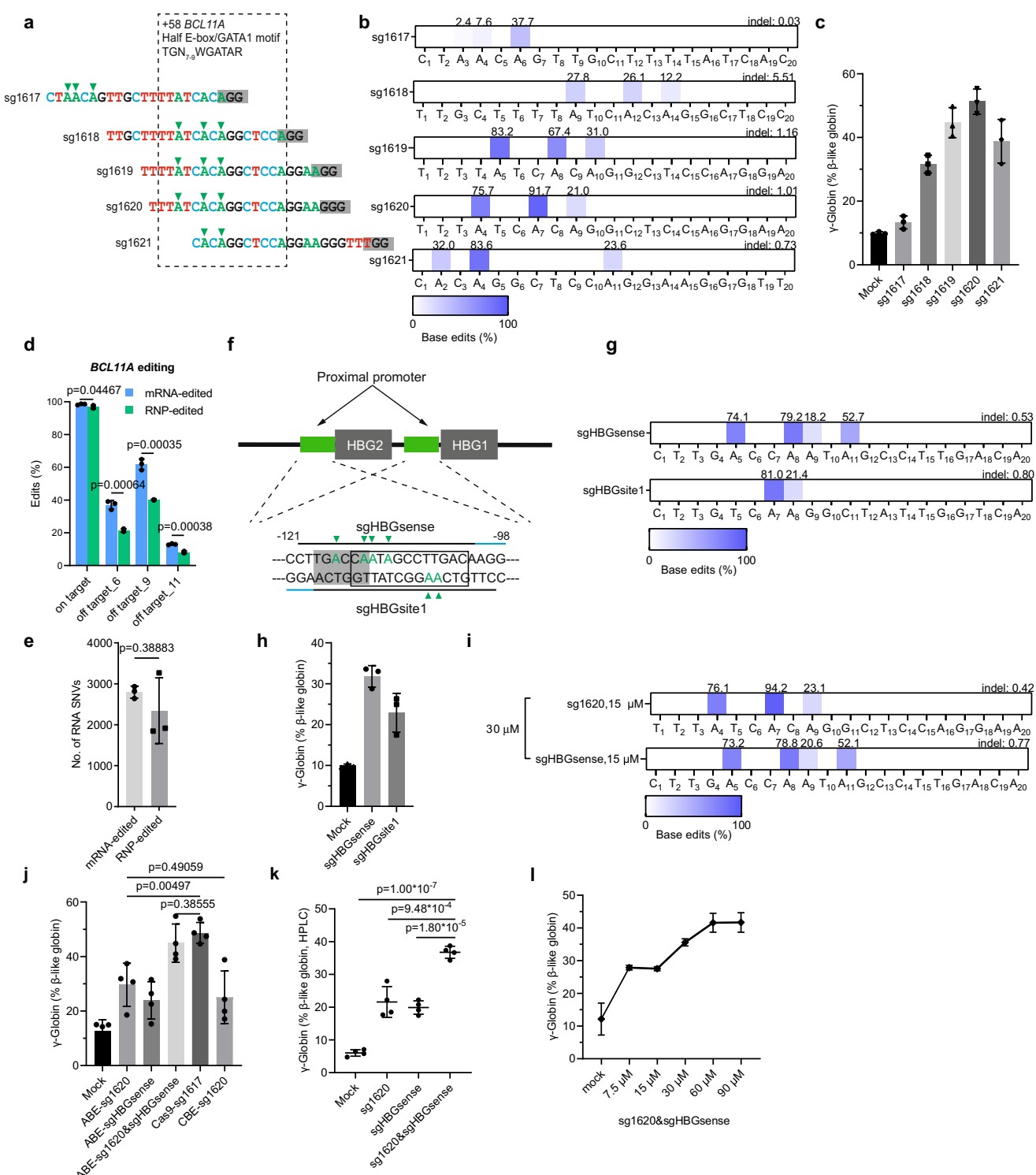

correlation between the base edit frequency at two target sites and the induced γ-globin mRNA level (Fig. 1l and Supplementary Fig. 5a). These data collectively show that ABE8e could benefit multiplex base editing applications in disruption of regulatory regions.

Previous studies have shown that both Cas9 and CBE can efficiently disrupt the +58 *BCL11A* erythroid enhancer[15,16] inducing γ-globin expression to a level that will be of clinical benefit to patients with β-hemoglobinopathies[27,28]. Here, for comparison, we also introduced highly efficient genome editing using 3 × NLS-SpCas9-sg1617 RNP and A3A (N57Q)-BE-sg1620 RNP, respectively[15,16] (Supplementary Fig. 5b). The ABE-sg1620, CBE-sg1620 and Cas9-sg1617 achieve target

modifications within their respective editing windows at the same GATA1 binding motif (Fig. 1a). RT-qPCR analysis of erythrocytes differentiated for 18 days in vitro showed that single editing by ABE8e resulted in γ-globin induction comparable to CBE and lower than Cas9 (Fig. 1j). Moreover, efficient multiplex editing with combined disruption of the *BCL11A* erythroid enhancer and the *HBG1/2* promoters mediated by ABE8e could increase the expression of γ-globin (44.90% of total β-like globin transcripts) as powerfully as Cas9 (Fig. 1j and Supplementary Fig. 4). These data above indicate that ABE8e, in addition to Cas9 and CBE, is able to effectively disrupt the *BCL11A* enhancer +58 locus to induce HbF level reaching or exceeding a clinical

**Fig. 1 | Efficient adenine base editing at the +58 *BCL11A* erythroid enhancer and *HBG* promoters in human CD34$^+$ HSPCs. a** Five sgRNAs targeting the core TGN$_{7-9}$WGATAR half E-box/GATA1 binding motif (shown in box) at +58 *BCL11A* erythroid enhancer with predominant base editing position indicated by arrowhead and PAM shaded. **b** Base editing by ABE8e complexed with each of five sgRNAs in human CD34$^+$ HSPCs. Heat map displays base edit frequency. **c**, **h** Percentage of γ-gene expression by RT-qPCR analysis after in vitro erythroid maturation of HSPCs edited by each of (or combined) the indicated sgRNAs complexed with ABE8e. Data are plotted as mean ± sd. $n = 3$ independent experiments. **d** Frequency of on-target and three validated sg1620-dependent DNA off-target sites with ABE8e mRNA and RNP delivery. Data were presented as mean ± sd, $n = 3$ independent experiments. **e** RNA off-target for mRNA and RNP groups. Data were presented as mean ± sd and analyzed with the unpaired two-tailed Student's *t* test. *p* values have been noted on the corresponding comparisons. $n = 3$ independent experiments. **f** Sequences of two sgRNAs targeting promoters of the *HBG*. BCL11A-

binding sites are shaded. The −114/−102 13 bp HPFH deletion is indicated by an empty box. Predominant base editing positions were indicated by green arrowheads. **g** Editing efficiency by ABE8e complexed with sgHBGsense or sgHBGsite1. **i** Multiplex base editing by ABE8e-sg1620&sgHBGsense. **j** Percentage of γ-gene expression determined by RT-qPCR in HSPCs edited by ABE8e-sg1620 or/and sgHBGsense, A3A (N57Q)-BE3-sg1620 and 3 × NLS-SpCas9-sg1617 targeting +58 *BCL11A* enhancer. Data were presented as mean ± sd. $n = 4$ independent experiments. **k** Percentage of γ-globin determined by RP-HPLC analysis after in vitro erythroid maturation of HSPCs edited by the sg1620, sgHBGsense, and sg1620&sgHBGsense complexed with ABE8e. Data are plotted as mean ± sd. $n = 4$ independent experiments. **l** γ-globin gene expression by RT-qPCR analysis of erythroid progeny after dose response of RNP electroporation of HSPCs. Data were presented as mean ± sd, $n = 3$ independent experiments. All statistical significance in the figures were analyzed with the unpaired two-tailed Student's *t* test, with *p* values noted where appropriate.

threshold (about 30% of HbF expression[29]) required to therapeutically ameliorate hemoglobin disorders[28].

## ABE8e editing is efficient and durable in healthy HSCs

We next investigated the editing efficiency in human repopulating HSCs. Mobilized CD34$^+$ HSPCs from one healthy donor were treated with ABE8e-sgRNA RNP complexes including ABE8e-sg1620, ABE8e-sgHBGsense, ABE8e-sg1620&sgHBGsense, respectively. Untreated and edited cells were injected into NCG-X (NOD-*Prkdc$^{em26Cd52}$Il2rg$^{em26Cd22}$kit$^{em1Cin(V831M)}$*/Gpt) immunodeficient mice. We assessed the editing efficiency and human haematopoietic lineages from isolated bone marrow (BM) after transplantation. The overall base editing frequency showed a similar modification profile in the input and in the engrafted human cells (Fig. 2a). Flow cytometry using an anti-human CD45 antibody showed that human cells made up more than 90% of BM in all mice (mean 97.0% for mock, mean 95.8% for sg1620 EP cell, mean 96.5% for sgHBGsense EP cells and mean 96.2% for sg1620&sgHBGsense EP cells, Fig. 2b). Flow cytometry to quantify the relative abundances of human myeloid (hCD33$^+$), B lymphoid (hCD19$^+$), and hCD19$^-$&hCD33$^-$ cells represented that the proportions of each lineage were roughly equivalent in mice that received unedited or edited cells (Fig. 2c and Supplementary Fig. 6), demonstrating that ABE8e treated normal HSPCs maintain capacity for engraftment and multilineage differentiation. RT-qPCR quantification of unfractionated BM cells revealed substantially elevated γ-globin mRNA levels from single edited groups compared to unedited control samples, accounting for 1.6% (unedited control), 8.2% (ABE8e-sg1620), and 10.8% (ABE8e-sgHBGsense) of total β-like globin transcripts in cells, respectively (Fig. 2d). Moreover, multiplex editing led to much higher γ-globin mRNA level (accounting for 20.3% of total β-like globin mRNAs) than single editing (Fig. 2d). Collectively, these findings indicate that base editing with ABE8e does not alter the engraftment or multipotency of transplanted HSCs.

We performed secondary transplantation to confirm that multilineage populating and self-renewal of long-term HSCs are not altered by adenine base editing. Maintained base edits was observed in secondary BM recipients (Supplementary Fig. 7a). Secondary transplantation revealed multilineage human hematopoietic engraftment in each secondary recipient consistent with HSC activity (Supplementary Fig. 7b, c). In unfractionated BM from co-editing group, γ-globin mRNA made up 10.8% (versus mean 1.5% in unedited group) of all β-like globins (Supplementary Fig. 7d). Together, these findings demonstrate that the ABE8e editing was durable in long-term haematopoietic stem cells.

## ABE8e mediates therapeutic adenine editing in thalassemia HSPCs

To further examine the potential benefit of base editing with respect to disease pathobiology, we then evaluated therapeutic HbF induction

by ABE8e RNP editing of primary HSPCs from patient with β-thalassemia. We electroporated ABE8e with sg1620, sgHBGsense, sg1620&sgHBGsense, respectively, into HSPCs from a patient with the β$^0$β$^+$ genotype (the donor is heterozygous for *HBB* codon 41/42 (-TCTT) together with *HBB* −28 (A > G), and hereafter termed β$^0$β$^+$ $_{\#1}$ patient). We observed efficient A•T to G•C conversion of sg1620-single editing (A$_4$: mean 81.3%, A$_7$: mean 94.3%, A$_9$: mean 24.5%) and sgHBGsense-single editing (A$_5$: mean 80.4%, A$_8$: mean 85.5%, A$_9$: mean 23.4%, A$_{11}$: mean 57.3%) in HSPCs from this patient, and multiplex editing also led to similar base edits at each target site as compared to single editing (Fig. 3a). RT-qPCR of globin genes showed increase in β-like globin relative to α-globin expression in erythroid progeny after therapeutic editing (Fig. 3b). Notably, co-editing had an additive effect on activation of γ-globin expression in comparison with single editing (Fig. 3b), as was the case with HSPCs from healthy donors (Fig. 1j, k and Supplementary Fig. 4).

Likewise, β$^0$β$^+$ $_{\#1}$ patient-derived HSPCs that had been untreated and co-edited were infused into NCG-X immunodeficient mice. After 16 weeks, we examined the functional potential of edited HSPCs. Editing efficiency at sg1620-target loci in the engrafted BM cells was virtually equal to the input cell populations whereas editing efficiency at sgHBGsense-target sequence was modestly decreased in BM compared with the input cells (Fig. 3c). Although there was lower human cell chimerism in co-edited HPSCs compared to nonelectroporated HSPCs (Fig. 3d), we observed similar profile of multilineage reconstitution between edited and unedited HSPCs (Fig. 3e). RT-qPCR analysis of globin mRNAs showed increased β-like globin to α-globin ratio balance in unfractionated BM after therapeutic combined editing, the ratio rising from 0.48:1 (unedited group) to 0.69:1 (co-edited group) (Fig. 3f). These findings indicate that the engraftment and differentiation potential of HSPCs from patient with β-thalassemia is not altered by ABE8e-mediated efficient base editing, providing a promising basis for TDT alleviating.

## PAM-relaxed ABE8e-SpRY enables in situ repair of *HBB* mutations

Induction of fetal hemoglobin (HbF) by ABE8e-mediated suppression and binding disruption of BCL11A has clinical promise, however, it may not fully suppress the expression of pathological β-globin. Elimination of the root cause of β-thalassemia by converting the *HBB* pathogenic allele to a normal or benign variant could overcome this limitation. However, the Cas9 nickase (D10A, nCas9) component of ABE8e is evolved from Streptococcus pyogenes (SpCas9) which naturally recognizes target sites with NGG protospacer adjacent motif (PAM). Thus, the PAM requirement is a major barrier for ABE8e application in correcting β-thalassemia pathogenic C·G → T·A variants such as two of the most prevalent β-globin gene mutations, HbE (CD 26, G > A) and IVS II-654 (C > T) that are not properly falling into the traditional base-editing window (typically, protospacer positions 4–8, counting the

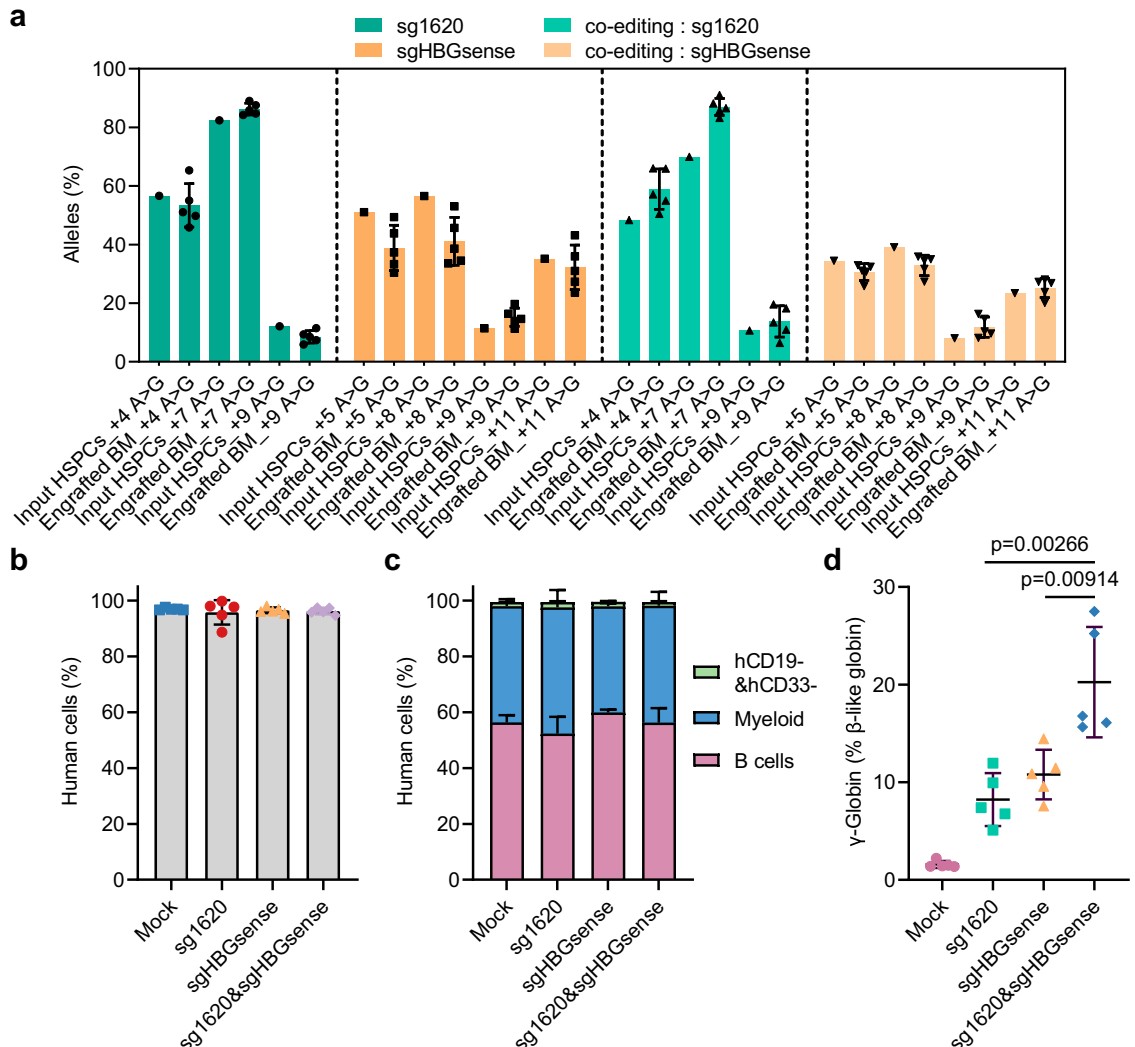

**Fig. 2 | Efficient A > G base editing in HSCs from healthy donor. a** Base editing in unfractionated BM after 16 weeks as compared to input HSPCs. Data are plotted as mean ± sd. *n* = 5 primary recipients for each group of edited HSPCs. **b** Comparing human chimerism of mock with all edited groups. Data are plotted as mean ± sd. *n* = 5 mice from mock and all edited groups. **c** Percentage of engrafted human B cells, myeloid cells and CD19⁻CD33⁻ cells 16 weeks after transplantation. Data were presented as mean ± sd, *n* = 5 mice. **d** γ-globin induction analyzed by RT-qPCR normalized by β-like globin, measured from BM chimerism 16 weeks after base edited HSPC infusion. Data are plotted as mean ± sd and analyzed with the unpaired two-tailed Student's *t* test, *p* values have been noted on the corresponding comparisons. *n* = 5 replicates from individual recipient mice.

PAM as positions 21–23)[19,22,30]. To establish the potential of ABE8e for widely expanded sequence targeting, we introduced 11 mutations (A61R/L1111R/D1135L/S1136W/G1218K/E1219Q/N1317R/A1322R/R1333P/R1335Q/T1337R) that were previously reported to support SpCas9 efficient targeting of many sites containing NRN and NYN PAMs[31] into its nCas9 domain to generate ABE8e-SpRY (Supplementary Fig. 1).

We next investigated whether delivery of ABE8e-SpRY into CD34⁺ cells can convert A to G in HSCs that are used to repopulate BM in an animal. At two test sites, RNP electroporation was performed with ABE8e-SpRY targeting *HEKsite4* (an endogenous genomic locus) guided by sgNGCT (GGGTCAGACGTCCAAAACCA, PAM: GGCT) and targeting *BCL11A* enhancer guided by sg1620 in CD34⁺ HSPCs from healthy donor. The overall base editing frequency in engrafted BM showed only a little reduction compared to input HSPCs (Supplementary Fig. 8a). There was similar human chimerism in all mice (mean 93.4% for mock, mean 91.7% for ABE8e-SpRY with sgNGCT and mean 95.1% for ABE8e-SpRY with sg1620; Supplementary Fig. 8b). Flow cytometry to quantify the relative abundances of specific lineages such as human B cells (CD19⁺), myeloid cells (CD33⁺) and hCD19⁻&hCD33⁻ cells showed that the proportions of each lineage were roughly

equivalent between control and ABE8e-SpRY-treated groups (Supplementary Fig. 8c). Gene expression in engrafted BM from sg1620-edited group demonstrated elevated γ-globin mRNA levels (mean 4.2% of total β-like globin, compared with 1.5% in sgNGCT-edited group and 1.5% in unedited group) (Supplementary Fig. 8), though the elevation of γ-globin expression should be further improved for potential therapeutic applications (Supplementary Fig. 8d). Since the results from these two test sites showed that ABE8e-SpRY could result in effective and durable A to G base editing in HSPCs, the therapeutic potential of ABE8e-SpRY-mediated in situ repair of pathogenic mutations (like HbE and IVS II-654) deserves to be evaluated.

HbE (CD 26, G > A) is a G > A substitution in codon 26 of the β-globin gene (*HBB*), which produces a structurally abnormal hemoglobin, hemoglobin E[32]. For HbE (CD 26, G > A) correction, ABE8e and ABE8e-SpRY with their respective sgRNAs were applied separately into CD34⁺ HSPCs from patients with Hb E/β-thalassemia (patients co-inherit a β-thalassemia allele from one parent and the structural variant hemoglobin E from the other, hereafter termed HbE patients). ABE8e guided by HbEsg1 could efficiently convert this mutant allele to wild-type allele in HSPCs (normal G percentage averaging 89.4% in edited

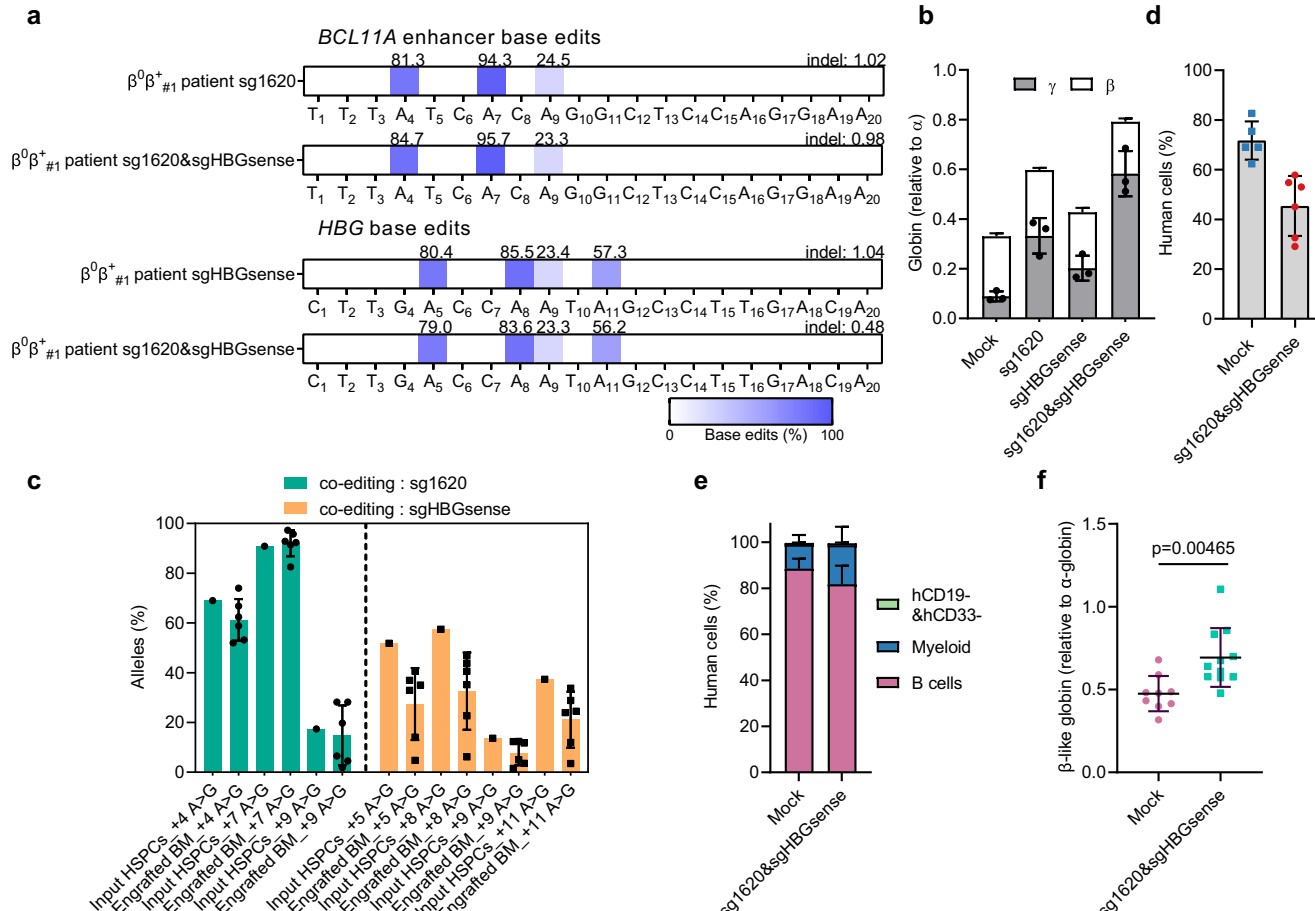

**Fig. 3 | Therapeutic and multiplex base editing in β-thalassemia patient CD34⁺ HSPCs. a** Base editing by ABE8e at +58 *BCL11A* enhancer with sgRNA-1620 (top two rows) and HBG promoter by sgHBGsense (bottom two rows) after single or multiplex editing in β-thalassemia donor (β⁰β⁺ #1 patient). **b** β-like globin expression by RT-qPCR normalized by α-globin, measured from edited erythroid progeny. Data are plotted as mean ± sd. $n = 3$ replicates from independent differentiation cultures. **c** Base editing in unfractionated BM after 16 weeks as compared to input HSPCs. Data are plotted as mean ± sd. $n = 6$ primary recipients for each group of engrafted HSPCs. **d** Human BM chimerism analyzation 16 weeks after base edited HSPC infusion. Data are plotted as mean ± sd. $n = 5$ mice for mock and $n = 6$ mice for edited group. **e** Percentage of engrafted human B cells, myeloid cells and CD19⁻CD33⁻ cells 16 weeks after transplantation (donor cells from β⁰β⁺ #1 patient). Data were presented as mean ± sd, $n = 5$ mice for mock and $n = 6$ mice for edited group. **f** β-like globin expression analyzed by RT-qPCR normalized by α globin, measured from β⁰β⁺ #1 patient donor BM chimerism 16 weeks after base edited HSPC infusion. Data are plotted as mean ± sd and analyzed with the unpaired two-tailed Student's *t* test, *p* values have been noted on the corresponding comparison. $n = 9$ replicates for mock, and $n = 11$ for edited group.

cells versus 50.1% in untreated cells) (Supplementary Fig. 9a). All five sgRNAs, in complex with ABE8e-SpRY, respectively, achieved mean normal G percentage ranging from 52.9% to 77.6%, with HbEsg2 behaved the highest editing frequency (mean 77.6%) (Supplementary Fig. 9a). Quantification of globin content following erythroid differentiation of edited human CD34⁺ cells showed both ABE8e&HbEsg1 and ABE8e-SpRY&HbEsg2 were sufficient to restore β-globin expression (Supplementary Fig. 9b–d). In addition, we observed minimal nonsynonymous bystander edits (less than 8.8% for ABE8e&HbEsg1 at an AGG PAM and less than 2.6% for ABE8e-SpRY&HbEsg2 at a CAGG PAM) as a result of careful positioning of the ABE[22,31] (Supplementary Fig. 10). Thus, introduction of ABE8e or ABE8e-SpRY using a clinically relevant delivery method can convert pathogenic A•T base pairs to non-pathogenic G•C in HSPCs efficiently and with few byproducts.

We then compared the editing performance of ABE8e and ABE8e-SpRY at the HbE site in engrafting HSPCs. The editing efficiencies of ABE8e and ABE8e-SpRY in input cells were 90.7% and 77.6%, while the mean editing efficiencies in engraft cells were 82.2% and 73.6%, respectively (Fig. 4a). Both ABE8e and ABE8e-SpRY-mediated grafts had similar proportions of human cells to the mock group, and edited HSPCs could differentiate into multiple lineages after transplantation

(Fig. 4b, c). The results of this trial provide validation that the engraftment and lineage survival of CD34⁺ cells were not altered by ABE8e-SpRY-mediated base editing. We analyzed the hemoglobin composition of CD235a⁺ cells in the BM of mice 16 weeks after transplantation using reversed-phase HPLC and found that repair of the HbE mutation by ABE8e-SpRY editing restored the proportion of normal β-hemoglobin to an average of 76.3%, even slightly higher than the average of 66.7% in the ABE8e-edited group (Fig. 4d). These results support the potential of both ABE8e and ABE8e-SpRY for therapeutically reversing pathogenic HbE codon 26 (G > A) mutation.

IVS II-654 (C > T) creates a de novo splice donor site in *HBB* intron-2, resulting in an aberrant β-globin mRNA containing an additional 73-nt exon that produces a premature stop codon[33]. However, there is no canonical NGG PAM neighboring the aberrant splice site necessary for ABE8e to correct the pathogenic mutation. To remove this constraint, we electroporated ABE8e-SpRY with each of five MS-sgRNAs (IVS654sg1-5) into human CD34⁺ HSPCs from patients carrying the IVS II-654 (C > T) heterozygous mutation. All five assessed RNP complexes mediated modest correction of IVS II-654 (C > T) to wild-type *HBB* genotype (the mean normal C percentage ranging from 57.2 to 77.9% in RNP-treated groups compared with 50.0% in untreated

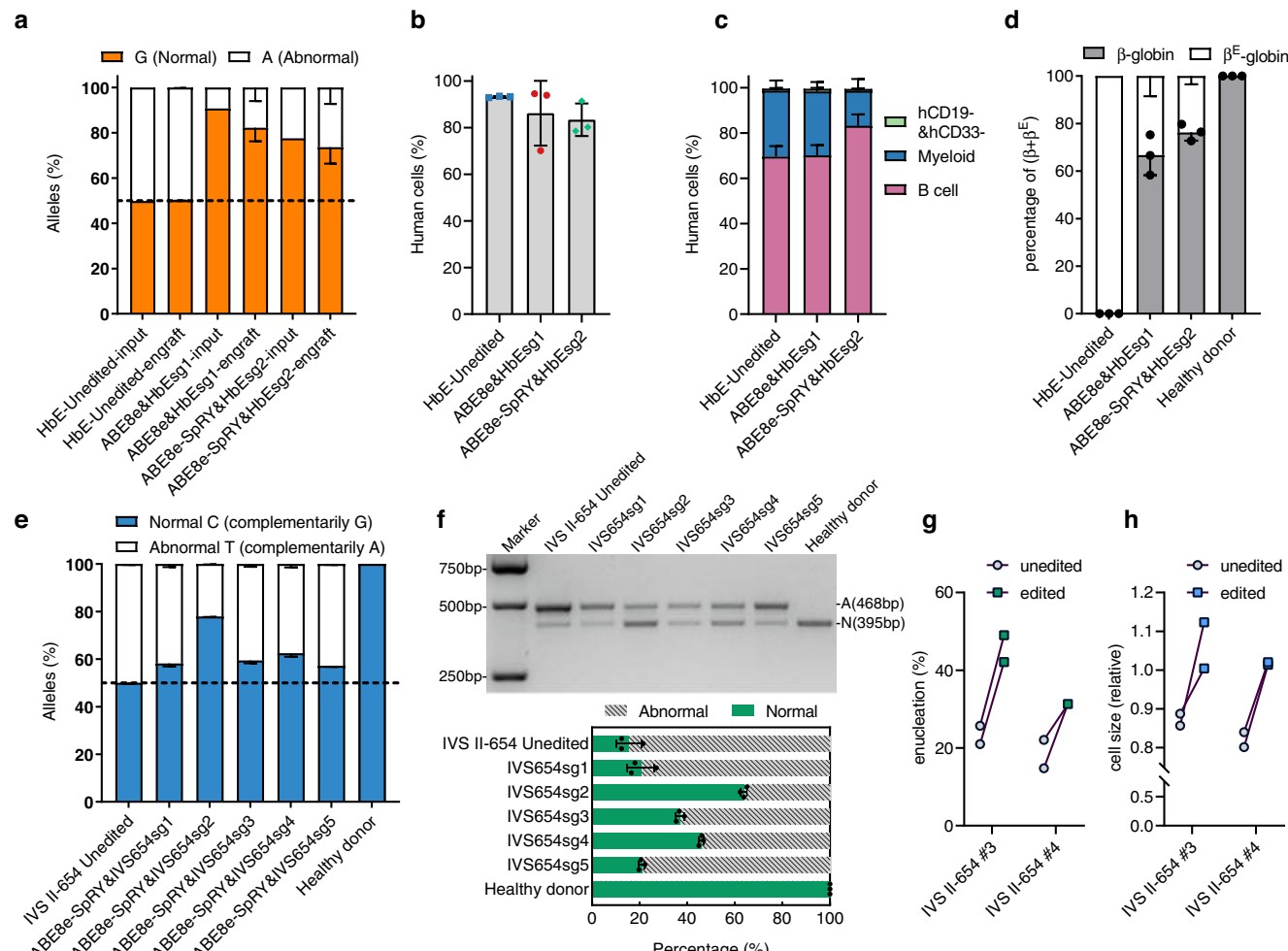

**Fig. 4 | ABE8e-SpRY therapeutic base editing in β hemoglobinopathy patients' CD34+ HSPCs. a** Target site allele frequency after ABE8e or ABE8e-SpRY editing in HbE/β-thalassemia patient's CD34+ HSPCs (HbE #3) and engrafts. Data are plotted as mean ± sd, n = 3 mice for each edited group. **b** The bone marrow chimeras were analyzed 16 weeks after transplantation, measured from grafts of HSPCs from HbE/β-thalassemia patient donors (HbE #3) edited by ABE8e targeted to HbEsg1 and ABE8e-SpRY targeted to HbEsg2 sites, respectively. Data are plotted as mean ± sd, n = 3 mice for mock and each edited group. **c** Percentage of engrafted human B cells, myeloid cells, and CD19⁻CD33⁻ cells, measured from grafts of HSPCs edited by ABE8e targeted to HbEsg1 and ABE8e-SpRY targeted to HbEsg2 sites, respectively. Data are plotted as mean ± sd, n = 3 mice for mock and each edited group. **d** RP-HPLC analysis of hemoglobin expression in mouse bone marrow CD235a⁺ cells transplanted with ABE8e or ABE8e-SpRY edited HbE/β-thalassemia patient cells (HbE #3). Data are plotted as mean ± sd, n = 3 mice. **e** Allele frequency of target

sites after ABE8e-SpRY editing in CD34⁺ HSPCs from patients (IVS II-654 #1-3) with IVS-II 654(C > T) mutation. Data are plotted as mean ± sd, n = 3 independent experiments. **f** RT-PCR from erythroid progeny with primers spanning the exon 2 to exon 3 junction, demonstrates abrogation of aberrant (A) and an increase in normal (N) splicing after therapeutic editing. Bands with a size of 468 bp were unrepaired cDNA PCR products, and bands with a size of 395 bp were normal (or repaired) cDNA PCR products. The below panel is the quantification of the above results (IVS II-654 #1-3). Data are plotted as mean ± sd, n = 3 experiments were repeated independently with similar results. **g** Enucleation of in vitro differentiated erythroid cells from 2 IVS II-654 donor HSPCs, each with two independent differentiations (IVS II-654 #3,4). **h** Cell size measured by relative forward scatter intensity for erythroid cells differentiated from two IVS II-654 donor HSPCs, each with two independent differentiations (IVS II-654 #3,4).

controls), with IVS654sg2 showed the highest normal C frequency (mean 77.9%) (Fig. 4e). To test whether genetic reversal of IVS II-654 (C > T) in CD34⁺ HSPCs is sufficient to restore β-globin splicing and expression, we performed gel electrophoresis (Fig. 4f). From a healthy donor sample, we observed only a single band of the expected size (395-bp amplicon). However, in the unedited patient samples, we observed an additional band demonstrating the expected aberrant splice product (468-bp amplicon). ABE8e-SpRY editing, with each of the 5 guide RNAs, reduced the aberrant splice product and reciprocally increased the normal splice product to different extents. Consistent with the deep sequencing data, we found the greatest normalization of the splice products for IVS654sg2, which reduced the percentage of aberrant splicing of β-hemoglobin mRNA from 84.5 to 36.3% (Fig. 4f). We hypothesized that therapeutical correction of IVS II-654 C > T mutation would result in improvement of terminal erythroid

maturation. Following ABE8e-SpRY editing, we observed a higher frequency of enucleation and larger size of differentiated erythroid cells from donors carrying this aberrant splice site mutation (Fig. 4g, h). These findings demonstrate that ABE8e-SpRY can be used to repair C•G > T•A mutations at sites without suitable canonical NGG PAMs, such as IVS II-654 (C > T), and provide proof-of-concept for the application of ABEs to mitigate inherited blood disorders caused by C•G > T•A mutations at extensive loci.

## Discussion

To mimic the positive effect on re-expressing γ-globin genes of the naturally occurring SNPs in *BCL11A* enhancer and to mimic mutations responsible for HPFH in the γ-globin gene promoters, in this study, we purified ABE8e for RNP electroporation of CD34⁺ HSPCs. We also described a bespoke ABE termed ABE8e-SpRY that directly corrected

the pathogenic HbE (CD 26, G > A) and IVS II-654 (C > T) mutations in the β-globin gene. This base editing strategy was efficient in HSPCs both in vitro and in vivo.

Approaches that have shown early clinical promise to treat TDT and SCD include addition of a normal β-globin coding sequence by lentiviral vectors[34] and induction of fetal hemoglobin (HbF) by Cas9-mediated disruption[3,5] or shRNA-mediated suppression[4] of *BCL11A*. Furthermore, the latter approach that reactivating the γ-globin gene by erythroid-specific knockdown of *BCL11A* is in early clinical development and achieved preliminary therapeutic progress[3–5]. It is not yet known which strategy is safest or most effective. However, the adenine base editing approach demonstrated here offers several potential advantages.

First, adenine base editing offers A to G base substitution with high product purity while bypassing the requirement for double-strand breaks (DSBs) or extrachromosomal template, thus largely avoids uncontrolled mixtures of indels at the target site as well as large deletions, chromosomal translocations, chromosome loss, chromothripsis, and activation of p53 in DNA damage response caused by DSBs[35–40].

Second, elimination of the disease-causing mutation by ABE-mediated precise editing may reduce the concentration of pathogenic hemoglobin more effectively than lentiviral expression of β-like globin or induction of HbF, both of which leave mutated *HBB* alleles intact.

We examined potential undesired consequences of base editing in HSPCs. Base editors can cause bystander editing of nearby nucleotides. In this study, ABE8e with HbEsg1 produced a small fraction of missense bystander alleles and ABE8e-SpRY with HbEsg2 produced even fewer byproducts (Supplementary Fig. 10).

Off-target base editing can also occur, although fewer off-target genome modifications were produced when ABE8e was delivered in RNP form than mRNA. Nevertheless, the safety and therapeutic potential of this approach might be further improved by adopting alternative adenine deaminase and Cas9 variants that have been shown to minimize deaminase-dependent and Cas-dependent off-target base editing[41], optimizing the dosage of the editing agent since on-target: off-target base editing ratios might be maximized by reducing exposure to base editor RNP[16] or optimizing delivery methods for less proneness to off-target genome modification.

Overall, our study provides proof that highly efficient, specific and disease-ameliorating adenine base editing in human HSCs is feasible with RNP delivery and supports the clinical development of ABE8e and ABE8e-SpRY editing in autologous HSCs as a potentially curative therapy for β-hemoglobinopathies.

## Methods

### Cell culture

Peripheral blood mobilized human CD34+ HSPCs from anonymous healthy donors were obtained from the First Affiliated Hospital of Zhejiang University School of Medicine (FAHZU), approved by the Medical Ethics Committee (MEC) of FAHZU, and informed consent was obtained from the donors. β-thalassemia patient CD34+ HSPCs were isolated from plerixafor-mobilized or unmobilized peripheral blood following Xiangya Hospital Central South University MEC, the First Affiliated Hospital of Guangxi Medical University MEC and PLA 923 Hospital MEC approval and informed patient consent. The donors were 7–9 years old. Genotype and other information of the cell donors involved in this study are listed in Supplementary Table 1. CD34+ HSPCs were enriched using the Miltenyi CD34 Microbead kit (Miltenyi Biotec) following manufacturer's instructions. CD34+ HSPCs were cryopreserved in CryoStor® CS10 (STEMCELL Technologies Inc.) and cultured into X-VIVO 15 (04-418Q, Lonza) supplemented with 100 ng ml$^{-1}$ human stem cell factor (SCF), 100 ng ml$^{-1}$ human thrombopoietin and 100 ng ml$^{-1}$ recombinant human Flt3-ligand. After 24 h of thawing,

HSPCs were electroporated with RNP. In vitro erythroid differentiation experiments were conducted 24 h after electroporation, HSPCs were transferred into erythroid differentiation medium (EDM) consisting of IMDM supplemented with 330 µg ml$^{-1}$ holo-human transferrin, 10 µg ml$^{-1}$ recombinant human insulin, 2 IU ml$^{-1}$ heparin, 5% human solvent detergent pooled plasma AB (Gemini), 3 IU ml$^{-1}$ erythropoietin, 1% L-glutamine and 1% penicillin/streptomycin. During days 0–7 of culture, EDM was further supplemented with $10^{-6}$ M hydrocortisone (Sigma), 100 ng ml$^{-1}$ human SCF and 5 ng ml$^{-1}$ human IL-3 (R&D) as EDM-1. During days 7–11 of culture, EDM was supplemented with 100 ng ml$^{-1}$ human SCF only as EDM-2. During days 11–18 of culture, EDM had no additional supplements as EDM-3. γ-globin induction was assessed on day 18 of erythroid culture.

### Animal handling

All experiments involving animals were performed according to the protocol approved by the ECNU Animal Care and Use Committee (protocol ID: m20200332) and in direct accordance with the Ministry of Science and Technology of the People's Republic of China on Animal Care Guidelines. Female 4–6 weeks NCG-X (NOD-Prkdc$_{em26Cd52}$Il2rg$_{em26Cd22}$kit$_{em1Cin(V831M)}$/Gpt) mice were ordered from GemPharmatech (Nanjing, China) (Stock T003802). In general, all mice were housed in 12:12 light: dark light cycles at room temperatures ranging between 20 and 26 °C and humidities between 30 and 70%. All mice were euthanized using carbon dioxide prior to collection of BM cells.

### Base editor protein expression and purification

ABE8e-6xHis tag, ABE8e-SpRY-6xHis tag, 6xHis tag-A3AN57Q and 3NLS-Cas9-6xHis (Addgene ID #114365) proteins, were expressed in E. Coli BL21 (DE3) (Thermo Fisher), which were grown in LB media at 37 °C and 4–5 h later induced by 1 mM isopropyl ß-d-1-thiogalactopyranoside for 18–20 h at 16 °C. Cells were collected and lysed by sonication in 500 mM NaCl, 20 mM Tris-HCl (pH 8.0), 1 mM TCEP and 10% glycerol buffer. The lysate was centrifuged at 10,000 × g for 45 min. The supernatant was filtered by 0.22 um Millex-GP Syringe filter unit (Millipore) and loaded onto HisTrap HP (GE). The proteins were eluted with a gradient of lysis buffer with 300 mM imidazole. The proteins were further dialyzed with SnakeSkin Dialysis Tubing (10 K MWCO, Thermo Fisher) for 24 h in the dialysis buffer containing 500 mM NaCl, 20 mM HEPES (7.5), 1 mM TCEP and 10% glycerol, enabling buffer exchange and low-molecular weight contaminant removal from sample solutions without significant loss of the macromolecule of interest. The dialyzed proteins were concentrated to ~15 mg/ml by Amicon ®Ultra-4 Centrifugal Filter Unit (Millipore) and stored at −80 °C.

### RNP and mRNA electroporation

Electroporation was performed using Lonza 4D Nucleofector (V4XP-3032 for 20 µl Nucleocuvette Strips or V4XP-3024 for 100 µl Nucleocuvette Strips) following the manufacturer's instructions. The chemical MS-sgRNA (2′-O-methyl 3′ phosphorothioate modifications in the first and last three nucleotides) was ordered from GenScript. CD34+ HSPCs were thawed and maintained in X-VIVO medium supplemented with cytokines 24 h before electroporation. For 20-µl Nucleocuvette Strips, the RNP complex was prepared by mixing protein (100 pmol) and sgRNA (300 pmol, full-length product reporting method) and incubating for 15 min at room temperature immediately before electroporation. Fifty thousand HSPCs resuspended in 20 µl of P3 solution were mixed with RNP and transferred to a cuvette for electroporation with program EO-100. For mRNA 20-µl Nucleocuvette Strips electroporation, the mRNA complex was prepared by mixing ABE8e mRNA (1 µg) and sgRNA (300 pmol). For 100-µl cuvette electroporation, the RNP complex was made by mixing 500 pmol ABE protein and

1500 pmol sgRNA. Next, 5 M HSPCs were resuspended in 100 µl of P3 solution for RNP electroporation as described above. The electroporated cells were resuspended with X-VIVO media with cytokines and changed into EDM 24 h later for in vitro differentiation. For mouse transplantation experiments, cells were maintained in X-VIVO 15 with SCF, TPO, and Flt3-L for 0–1 day as indicated before infusion. Single guide RNA sequences, PAM and related deep-seq primers are listed in Supplementary Data 1.

## Base editing results measurement
Cells cultured in EDM 96 h after electroporation were subjected to edit frequencies measurement. Briefly, genomic DNA was extracted using the TIANamp Micro DNA Kit or TIANamp Genomic DNA Kit. *BCL11A* enhancer DHS + 58 core and *HBG1/2* promoters −115 region were amplified with KOD-Plus-Neo DNA Polymerase and corresponding primers using the following cycling conditions: 95 °C for 3 min; 35 cycles of 98 °C for 10 s, 60 °C for 30 s and 68 °C for 15 s; and 68 °C for 5 min. Resulting PCR products were subjected to Sanger sequencing or Illumina deep sequencing. For Sanger sequencing, traces were imported to EditR software[42] for base editing measurement. For deep sequencing, *BCL11A* enhancer loci or *HBG1/2* promoters loci were first amplified with corresponding primers. After another round of PCR with a pair of site-specific primers with common bridging sequences added at the 5′ end[43], amplicons were sequenced for 2 × 150 paired-end reads with the MiSeq Sequencing System (Illumina). Frequencies of editing outcomes were quantified using CRISPResso2 software[44] (version 2.1.3, CRISPRessoBatch -quantification_window_center −10 -quantification_window_size 10 -base_editor_target A -base_editor_result G -base_editor_output TRUE) and collapsed on the basis of mutations in the quantification window. Indels overlapping the spacer sequence were counted as indels, and A > G substitutions at spacer positions 1–10 were counted as base edits for total edit quantification. For base edit heat maps, indels were excluded before calculation of nucleotide substitution frequency.

## Guide-RNA-dependent off-target editing analysis
Twenty-eight potential off-target sites based on sg1620 and 10 potential off-target sites based on sgHBGsense, respectively, with three or fewer genomic mismatches and no bulges were identified using the CasOFFinder tool[45]. The detection primers for these sites were designed using NCBI primer -blast tool and synthesized by GENEWIZ (Suzhou, China). Fifty nanograms of genomic DNA was used per reaction using the following cycling conditions: 95 °C for 3 min; 35 cycles of 98 °C for 10 s, 60 °C for 30 s and 68 °C for 15 s; and 68 °C for 5 min. Amplicons were sequenced in the MiniSeq Sequencing System, and analyzed in CRISPResso (version 2.1.3, CRISPRessoBatch -quantification_window_center −10-quantification_window_size 10-base_editor_target A -base_editor_result G -base_editor_output TRUE). One-tailed Student's *t* tests (α = 0.05) were used to compare mean editing frequencies between edited and unedited samples for each target site. Potential off-target sites with editing frequency difference between control and edited samples of at least 0.1% and with $p < 0.05$ were considered as confirmed off-target sites. Measured edits greater than 0.1% were visually inspected to evaluate for potential sequencing or alignment artifacts. 3 sg1620 off-target sites OFT6, OFT9 and OFT11 were compared to ATAC-seq peaks[46] and with conservation scores using the UCSC PhastCons track on the basis of multiple alignments of 100 vertebrate genomes to the human genome[47]. These sites did not overlap coding sequences or chromatin-accessible peaks and showed very low conservation scores [OFT6(0, 0, 0), OFT9(0, 0.00496063) and OFT11(0.165354, 0.141732, 0, 0), each value for an off-targeting base A in the editing window]. Predicted off-target site information of sg1620 and sgHBGsense and related primers are listed in Supplementary Data 2 and 3.

## Hemoglobin RP-HPLC
Hemolysates were prepared from erythroid cells after 18 days of erythroid differentiation for in vitro differentiation experiments and reverse phase HPLC was performed on an Agilent 1260 infinity II using the 4.6-nm Aeris 3.6 mM Widepore C4 LC column.

## RT-qPCR quantification of globin expression
RNA isolation with RNAsimple Total RNA Kit (DP-419, TIANGEN), reverse transcription with ReverTra Ace™ qPCR RTMaster Mix with gDNA Remover Kit (FSQ-301, Bio-Rad) and RT-qPCR with FastStart Universal SYBR Green Master (Rox) (04 913 914 001, Roche) were used to determine globin expression with primers amplifying *HBG1/2*, *HBB* or *HBA1/2* cDNA[15]. All gene expression data represent the mean of at least three technical replicates, with biological replicates annotated in the corresponding legends.

## Human CD34⁺ HSPC transplant and flow cytometry analysis
All animal experiments were approved by University Committee on Animal Research Protection of East China Normal University. We had complied with all relevant ethical regulations. CD34⁺ HSPCs were obtained from anonymized healthy donors or from β-hemoglobinopathy patients. NCG-X (NOD-*Prkdc^{em26Cd52}Il2rg^{em26Cd22}ki-t^{em1Cin(V831M)}*/Gpt) mice were ordered from GemPharmatech (Nanjing, China) (Stock T003802). Nonirradiated NCG-X female mice (4–6 weeks of age) were infused by retro-orbital injection with 0.8 M CD34⁺ HSPCs (live cells counted immediately before infusion, resuspended in 200 µl of DPBS) derived from healthy donors or β-hemoglobinopathy patients. BM was isolated for human xenograft analysis 16 weeks after engraftment. Secondary transplants were conducted using retro-orbital injection of BM cells from the primary recipients. For flow cytometry analysis of BM, BM cells were first incubated with Human TruStain FcX (422302, BioLegend) and TruStain fcX (anti-mouse CD16/32, 101320, BioLegend) blocking antibodies for 10 min, followed by incubation with Brilliant Violet 421™ anti-human CD45 Antibody (304032, Biolegend), PE/Dazzle™ 594 anti-mouse CD45 Antibody (103146, Biolegend), PE anti-human CD235a (Glycophorin A) Antibody (349106, Biolegend), FITC anti-human CD33 Antibody (303304, BioLegend), APC anti-human CD19 Antibody (302212, BioLegend), and Fixable Viability Dye eFluor 780 for live/dead staining (65-0865-14, Thermo Fisher). Percentage human engraftment was calculated as hCD45⁺ cells/ (hCD45⁺ cells add mCD45⁺ cells) × 100. B cells (CD19⁺) were gated on the hCD45⁺ population. Granulocytes and monocytes were gated on the hCD45⁺hCD19⁻ population. Human erythroid cells (CD235a⁺) were gated on the mCD45⁻hCD45⁻ population.

## RNA sequencing and SNV calling
Total RNA was isolated and purified from electroporated and mock cells using TRIzol reagent (Invitrogen, CA, USA) following the manufacturer's procedure. After testing the purity and integrity of the RNA, mRNA sequencing library was constructed following the standard protocol. Sequencing was performed on the Illumina HiSeq X Ten platform with PE150 strategy. For RNA-seq data analysis, FastQC (v0.11.9) and cutadapt (v3.4) were used for sequencing quality control and adaptor removing. Qualified reads were mapped to the reference genome (Ensemble GRCh38.p13) using STAR (v2.7.9a) in 2-pass mode with the parameters implemented by the ENCODE project. Sambamba (0.8.1) was then applied to mark and remove PCR duplicates of the mapped BAM files. The refined BAM files were subject to split reads that spanned splice junctions, base quality recalibration and variant calling with SplitNCigarReads, BaseRecalibrator, and HaplotypeCaller tools from GATK (v4.2.1.0) respectively. To enhance the confidence of the SNVs found, we filtered them using the following criteria: (1) filtered clusters of at least 5 SNVs that were within a window of 35 bases; (2) filtered variants with base-quality score <25.0, sequencing depth

<20, mapping quality score <40.0, Fisher strand values >30.0, qual by depth values <2.0, MQRankSum <−4.0 and ReadPosRankSum <−4.0. We counted the sum of A to G + T to C for ABE editing rate, as the mRNAs were converted into cDNA before sequencing, and both the nucleotide and its complementary base could be sequenced. Confident variants found in mock group cells were considered to be SNPs and were removed from the RNP-edited and mRNA-edited groups for off-target analysis with bcftools (v1.13). The editing rate was calculated as the number of mutated reads divided by the sequencing depth for each site.

## Statistics and reproducibility

Statistical analyses were performed using Prism 8.0 (GraphPad Software). No sample size calculation was performed in this article. The sample sizes for all statistical comparisons were made using the community default criteria, i.e., biological replicates ≥3. No data were excluded from the analyses. Data obtained from edited input cells for transplantation experiments were successfully performed with more than three technical replicates. The cells were then divided equally and transplanted into multiple recipient mice. All the other experiments are replicated more than three times. All attempts at replication were successful. Samples were allocated into experimental groups randomly. The Investigators were not blinded to allocation during experiments and outcome assessment.

## Reporting summary

Further information on research design is available in the Nature Portfolio Reporting Summary linked to this article.

## Data availability

All amplicon deep-sequencing data and RNA sequencing data generated in this article can be found at the National Center for Biotechnology Information's Sequence Read Archive with accession code PRJNA892449. All data supporting the findings of this study are available within the article and Supplementary Information files and also are available from the corresponding author upon request. Source data are provided with this paper.

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

## Acknowledgements

This work was partially supported by grants from the National Key R&D Program of China 2019YFA0109901 (Y.W.), and 2019YFA0110803 (Y.W.), grants from the Shanghai Municipal Commission for Science and Technology 19PJ1403500 (Y.W.), the National Science Foundation of China grants 32001061 (S.C.), China Postdoctoral Science Foundation grants 2019M661430 (S.C.), 2019TQ0096 (S.C.), 2020M681231 (J.L.). We thank Ying Zhang from the Flow Cytometry Core Facility of School of Life Sciences of ECNU for help on flow cytometry analysis data collection.

## Author contributions

J.L. and S.C. designed and performed experiments. S.H. and Y.J. helped with experiments conduction and data collection. Y.Z. and X.W. helped with RP-HPLC experiment. J.L. performed computational analyses, oversaw data analysis/interpretation. Y.Y. and Y.L. collected and genotyped cells from patients, provided advice on cytomorphological measurements. Y.W. and S.C. initiated the project, supervised the research. D.E.B. contributed to the final version of manuscript. S.C., J.L. and Y.W. analyzed experiments and wrote the manuscript with input from all authors.

## Competing interests

The authors declare no competing interests.
