## [Peer Review File · Nature Communications]

Therapeutic adenine base editing of human hematopoietic stem cellsReviewer #1 (Remarks to the Author):

Verdict: Revise and resubmit

Chen and co-authors apply adenine base editors as a potential therapeutic for β -thalassemia. The authors claim:

- 1. The evolved adenine base editor, ABE8e, generates A to G nucleotide substitutions in the BCL11A enhancer and HBG1/2 promoter regions.**
- 2. Ex vivo delivery of ABE8e complexed with sgRNAs targeting these regions results in reactivation of fetal hemoglobin (HbF) in both healthy and patient HSPCs.**
- 3. Simultaneous editing of the BCL11A enhancer and HBG1/2 promoters yields increased HbF stimulation compared to targeting either region alone.**
- 4. Development of ABE8e-SpRY, which is nearly PAM non-specific, to correct two β -thalassemia mutations, HbE and IVS2-654.**
- 5. Recovered ABE8e edited human HSCs displayed high editing efficiency in primary and secondary recipient animals, where γ -globin and normal phenotypic multilineage repopulation was observed.**

The manuscript clearly shows that ABE8e can be optimized to perform as well as Cas9 biallelic editing. Multiplex editing of the BCL11A enhancer and the HBG1/2 promoter, with ABE8e, induces a striking \sim 4-fold gain in HBG expression, with similar parity to Cas9 editing of the BCL11A enhancer alone. However, the authors are using reagents and workflows produced and validated by other labs, swapping only the editing component. Therefore, this finding alone reads as an incremental advance.

The concept of a promiscuous "near-PAMless" ABE is intriguing and novel, promising potential access to the entire genome. However, the efficacy of the ABE8e-SpRY construct remains in question. HbE editing rates compared to ABE8e, even when using the same sgRNA, are not nearly as robust, while HBB rescue in the context of IVS2-654 is low. ABE8e-SpRY induced HBG stimulation is low or non-existent. Furthermore, the depth of data on ABE8e-SpRY in comparison to ABE8e is not as complete. It is not possible in the current manuscript to evaluate whether or not ABE8e-SpRY has therapeutic value.

Furthermore, the manuscript has a number of errors and inconsistencies that made it challenging to read and evaluate. Major and minor concerns are listed below.

Major Concerns

- 1. The lack of reported effect size in the manuscript obscures significance and therapeutic potential. Specifically, the authors should define what is a therapeutic threshold, or at least make a reasonable approximation thereof. Additionally, the authors should use concise definitions of the size of an effect, not ambiguous terms like "efficient" and "potent". Specific examples of this include:
 - i. Line 114 – "potently". Define in fold changes.**
 - ii. Line 125 – "substantially increased". Define in fold changes.**
 - iii. Line 148 – What is clinical threshold?**
 - iv. There are many other examples throughout the text and abstract.****
- 2. The inconsistent use of terms and assays makes comparison between or even within figures challenging.
 - a. The names of editing reagents changes between figures, i.e. ABE8e-sg1620/sghBGsense in Figure 1 and BCL11A/HBG in Figure 2.**
 - b. Figure 1j and 1m are similar assays but use different x-axis nomenclature.**
 - c. Globin expression measurements appear to change between figures. Figure 1 only measures HBG, whereas subsequent figures mention both HBG and beta-like globin. Authors should only report HBG levels as this can be directly attributed to their editing, exempting the correction of HbE and IVS2-654 where HBB should be used.****
- 3. The authors' ABE8e-SpRY construct does not seem to work well. Editing efficiency is**

low, significantly less than ABE8e. There is low correction of HBB in the context of HbE and IVS2-654. In extended data figure 10, there is minimal change in beta like globin expression. Finally, there is a dearth of primary and secondary engraftment data of patient cells edited for HbE and IVS2-654, which seems to have been substituted for engraftment data of healthy HSPCs using sgRNA1620 and sgNGCT.

4. There are general issues with regards to data quality and formatting.

a. Some error bars appear to be too narrow given data in the manuscript. Figures 1c and 1j appear to use the same reagents and conditions, but the editing rates are much lower in 1j than 1c. If these were edited the same way, they should have been pooled and the error bars should be revised.

b. Some panels contain redundant data and should be combined. Specifically, Figure 1j and 1m appear to have the same data in the first four bar graphs, where the axis labels are changed in the latter panel.

Minor Concerns

1. Line 292- This is not single base substitution. There are varying probabilities that 3 sites are changed.

2. Figure references in the text are wrong, for example:

a. Lines 113-118 – figures references are wrong, they should be f-h.

3. Figure legends are incorrect, to list a few instances:

a. Fig 4e = mean +/- s.e.m. of how many?

b. Many typos in Figure 4 legend.

c. Fig 4e legend does not define which site is which.

4. There are statements in the manuscript that are inconsistent with data in the manuscript.

a. It is not interesting that the ABE8e-SpRY variant doesn't have high indel levels if the overall editing rates are lower as well.

b. Disagree with statement 239. The authors do not convince me that the SpRY construct has therapeutic potential.

c. Disagree with statement 259. The authors own data show that there is minimal if any increase in HBG and the term "globin chain balance" is nebulous and should be properly defined.

5. There is a dire loss of cells in double edited secondary recipients in Extended data Fig 7b.

a. Is this due to HBG editing? Double edits? What do the authors think is happening here?

Reviewer #2 (Remarks to the Author):

This manuscript by S. Chen reports progress in adapting the adenine base editor, ABE8e, for genomic alterations that induce fetal hemoglobin and correct selected β -thalassemia mutations. Together these strategies might be applied in clinically significant β -hemoglobinopathies. The first series of experiments convincingly show successful targeting of the erythroid-specific enhancer of BCL11a and HGB 1/2 promoters by optimized sgRNA complexed in an ABE8e RNP that generated high level editing and induced γ -globin expression both in vitro and in xenografted mice from human HSPCs. While neither these genomic targets for γ -globin induction by genomic editing is novel, this represents incremental progress in developing safe and effective editing reagents. While there also appears to be an additive effect by multiplex editing, this also has the potential to amplify off-target and other deleterious effects of editing. In addition, other published evidence strongly suggests that editing at a single target locus (either BCL11a or HGB 1/2) is sufficient for mitigating phenotypic expression of β -hemoglobinopathies.

The strength of the report is the progress shown in adapting ABE8e to the near PAM-less variant, ABE8e-SpRY, for which sgRNA optimization was executed to target HbE and IVS2-654 thalassemia mutations. Although, it is not yet clear if the correction rate demonstrated in the manuscript is sufficient, the possibility of using base editors in lieu of inefficient template-directed homologous repair of common β -hemoglobinopathy mutations by Cas RNPs is both significant and very interesting, and has the potential to advance the field.

Comments:

1. I do not agree that BCL11A and HBG promoters can be viewed as having independent regulatory responses in the context of these experiments (Line 119 of ms). Both the targeted HBG and BCL11a genomic modifications were intended to inhibit γ -globin transcription repression by BCL11a. These are not independent actions. If the editing at both loci had independent mechanistic effects that were interactive, the multiplex editing should have generated a cooperative effect on γ -globin de-repression, which it did not. At best, the multiplex effect was additive.
2. The xenotransplantation mouse strain used in these experiments (NCG-X) permitted remarkably robust human hematopoietic cell engraftment, exceeding 95% (Fig 2b). Is it possible that this level of engraftment generated anemia in the mouse at 16 weeks? If so, murine paracrine signaling could have a cross-reactive effect on human erythropoiesis in the mouse, and contribute to HbF induction in human erythroid progenitor cells. Was this possibility investigated? It might be important to repeat some of the xenografting studies in NSG mice where engraftment is not as efficient.
3. The assessments of γ -globin expression (as a proportion of total β -like globin expression) in Fig 2d and Ext data Fig 7 should use the same scale to facilitate comparison of expression in primary and secondary xenografts. The secondary mice appear to show a smaller contribution in γ -globin expression, which could indicate a problem with HSC propagation.
4. Based on the data in Fig 3b (with HSPCs from a $\beta^0\beta^+$ thalassemia donor) and presented in Line 205, it is important to note that improving the β -globin to α -globin ratio from 0.5 to 0.7 is not sufficient to overcome all the deleterious effects of globin chain imbalance and ineffective erythropoiesis.
5. The HSPCs from the E/ β thal donor appeared to have no β -like globin expressed in unedited cells (Fig 4b). HbE/ β thal disease is very heterogeneous, according in part to whether there is β^0 or β^+ allele. The genotype of the HbE donor was not reported, and ideally HSPCs from several patients should be analyzed. The same would be true for HSPC donors with the IVS2-654 mutation.
6. The Fig 4 heading (...therapeutic editing in β -thalassemia/HbE patient...) also presents data IVS2-654 splice junction editing. The heading should be corrected to include both genotypes.
7. The paragraph in the discussion that begins at line 308 reports primary off-target data, which should be moved to the results section and referred to in the discussion.
8. Some of the xenotransplantation studies used just 3 mice, others 5, but in every group the variability was reported as SEM in lieu of standard deviation. The standard deviation more rigorously tests for statistical significance and should be used to assign significance in comparisons.

Dear reviewers,

We thank the reviewers for their critical evaluation and comments on our manuscript, which were important in improving our manuscript (# NCOMMS-21-41175).

Before our point-to-point responses, an important background information needs to be introduced as follows:

Since the efficacy and therapeutic value of ABE8e-SpRY is focus of our attention, we did additional efficiency studies and discovered that sgRNAs supplied by different providers had varying editing efficiency. As shown in figure below, sgRNAs produced by GenScript utilizing different purification procedures can increase the editing efficiency of the IVS654sg2 site from 42.5% (Syn, Synthego sgRNA) to 55.9% (Gen, Genscript sgRNA). Meanwhile, subsequent electroporation(2EP) can boost target editing efficiency to 79.3%. Similarly, the editing efficiency and therapeutic effect of different batches of sgRNAs targeting the HbE site were also compared. For ABE8e&HbEsg1, the sgRNA provided by GenScript(Gen) has slightly improved editing efficiency and editing effect compared with Synthego(Syn), the editing efficiency has increased from 76.9% to 78.8%, and the proportion of β -globin in erythroid cells from in vitro differentiation has increased from 76.0% to 82.3%. For ABE8e-SpRY&HbEsg2, there was a significant improvement, the editing efficiency increased from 23.3% to 55.2%, and the proportion of β -globin in erythroid cells from in vitro differentiation increased from 20.0% to 57.5%. These data imply that the editing efficiency of ABE8e-SpRY will be sufficiently enhanced if the sgRNA synthesis and purification processes, as well as the electroporation process, are optimized.

We found only slight improvements in GenScript-synthesized sgRNAs at relatively high editing efficiencies (~higher than 75%) situation, but improvements at fairly low editing efficiencies situation were more pronounced. Therefore, we used GenScript-synthesized sgRNA to refine the study in the experiment of ABE8e-SpRY editing to repair in situ mutations (HbE & IVS II-654). The therapeutic effect of editing has been considerably enhanced by the change in editing efficiency (see figure 4 for more information), revealing the therapeutic application potential of ABE8e-SpRY.

The detailed point-to-point responses are as follows:

Responses to Reviewer #1

Reviewer #1 (Remarks to the Author):

Verdict: Revise and resubmit

Chen and co-authors apply adenine base editors as a potential therapeutic for β -thalassemia.

The authors claim:

1. The evolved adenine base editor, ABE8e, generates A to G nucleotide substitutions in the BCL11A enhancer and HBG1/2 promoter regions.
2. Ex vivo delivery of ABE8e complexed with sgRNAs targeting these regions results in reactivation of fetal hemoglobin (HbF) in both healthy and patient HSPCs.
3. Simultaneous editing of the BCL11A enhancer and HBG1/2 promoters yields increased HbF stimulation compared to targeting either region alone.
4. Development of ABE8e-SpRY, which is nearly PAM non-specific, to correct two β -thalassemia mutations, HbE and IVS2-654.
5. Recovered ABE8e edited human HSCs displayed high editing efficiency in primary and secondary recipient animals, where γ -globin and normal phenotypic multilineage repopulation was observed.

The manuscript clearly shows that ABE8e can be optimized to perform as well as Cas9 biallelic editing. Multiplex editing of the BCL11A enhancer and the HBG1/2 promoter, with ABE8e, induces a striking \sim 4-fold gain in HBG expression, with similar parity to Cas9 editing of the BCL11A enhancer alone. However, the authors are using reagents and workflows produced and validated by other labs, swapping only the editing component. Therefore, this finding alone reads as an incremental advance.

The concept of a promiscuous “near-PAMless” ABE is intriguing and novel, promising potential access to the entire genome. However, the efficacy of the ABE8e-SpRY construct remains in question. HbE editing rates compared to ABE8e, even when using the same sgRNA, are not nearly as robust, while HBB rescue in the context of IVS2-654 is low. ABE8e-SpRY induced HBG stimulation is low or non-existent. Furthermore, the depth of data on ABE8e-SpRY in comparison to ABE8e is not as complete. It is not possible in the current manuscript to evaluate whether or not ABE8e-SpRY has therapeutic value.

Furthermore, the manuscript has a number of errors and inconsistencies that made it challenging to read and evaluate. Major and minor concerns are listed below.

Thank you for reviewing this manuscript and making such insightful suggestions. A detailed response to your concerns is attached below.

Major Concerns

1. The lack of reported effect size in the manuscript obscures significance and therapeutic potential. Specifically, the authors should define what is a therapeutic threshold, or at least make a reasonable approximation thereof. Additionally, the authors should use concise definitions of the size of an effect, not ambiguous terms like “efficient” and “potent”. Specific examples of this include:
 - i. Line 114 – “potently”. Define in fold changes.

- ii. Line 125 – “substantially increased”. Define in fold changes.
- iii. Line 148 – What is clinical threshold?
- iv. There are many other examples throughout the text and abstract.

We apologize for the confusion caused by the use of ambiguous terms to describe the experimental results. And 30% of HbF expression would be considered a significant curative threshold able to prevent α free chains polymerization in β -thalassemia and HbS precipitation in SCD. We rechecked the manuscript to ensure that vague descriptions had been revised.

2. The inconsistent use of terms and assays makes comparison between or even within figures challenging.

a. The names of editing reagents changes between figures, i.e., ABE8e-sg1620/sgHBGsense in Figure 1 and BCL11A/HBG in Figure 2.

b. Figure 1j and 1m are similar assays but use different x-axis nomenclature.

c. Globin expression measurements appear to change between figures. Figure 1 only measures HBG, whereas subsequent figures mention both HBG and beta-like globin. Authors should only report HBG levels as this can be directly attributed to their editing, exempting the correction of HbE and IVS2-654 where HBB should be used.

2a. Thanks to the reviewer's reminder, we revised the manuscript related content and unified the name of the editing reagent.

2b. We removed redundant data and redirected the corresponding description in the text to the same figure.

2c. We would like to provide some explanations why the induction of γ -globin is shown differently. For healthy human donor cells, we used $\gamma/(\gamma+\beta)$ to demonstrate the induction of γ globin after editing. This was calculated with a significant percentage of β - globin in healthy donor cells. As described by the reviewer, this allows direct characterization of the induction effect of HBG. However, the proportion of β -globin in the donor cells of patients with β -thalassemia is low or absent, and using the calculation method of $\gamma/(\gamma+\beta)$ will make the results of the editing group and the control group approach 1 and cannot be distinguished. Therefore, for β -thalassemia patient donor cells, the $(\gamma+\beta)/\alpha$ calculation method is generally used, and this expression method can intuitively show the degree of balance of the globin chain in the blood of the thalassemia patient after editing.

3. The authors' ABE8e-SpRY construct does not seem to work well. Editing efficiency is low, significantly less than ABE8e. There is low correction of HBB in the context of HbE and IVS2-654. In extended data figure 10, there is minimal change in beta like globin expression. Finally, there is a dearth of primary and secondary engraftment data of patient cells edited for HbE and IVS2-654, which seems to have been substituted for engraftment data of healthy HSPCs using sgRNA1620 and sgNGCT.

As shown in fig4a, d of the revised manuscript, after the use of the new sgRNA, the editing efficiency of ABE8e-SpRY to repair the HbE mutation, as well as the degree of correction of β^E , has approached that of ABE8e. Besides, the advantage of ABE8e-SpRY is that there is almost

no editing site restriction, thus multiple sgRNAs can be tested and the one with relatively high editing efficiency can be selected.

The data presented in the original manuscript is a collection of experimental results of sgRNAs from different suppliers. When compared separately, we found that Genscript sg was significantly more efficient and effective than Synthego sg.

After ABE8e-SpRY & Sg1620 editing, the γ induction effect is not significant enough because the sg1620 locus is not necessarily suitable for ABE8e-SpRY. This is a test site to detect whether ABE8e-SpRY can edit long-term HSCs. Cell transplantation after sgRNA1620 and sgNGCT loci editing could demonstrate that ABE8e-SpRY could achieve editing of long-term HSCs.

We recently completed the primary transplantation of HbE cells edited by ABE8e-SpRY with the results in figure 4a-d, which demonstrated that ABE8e-SpRY had comparable therapeutic potential to ABE8e at the HbE site. For IVS2-654, transplantation could not be performed due to the lack of sufficient number of donor cells. We still tried our best to finish in vitro differentiation assays. ABE8e-SpRY editing, with IVS654sg2, reduced the percentage of aberrant splicing of β -hemoglobin mRNA from 87.6% to 35.0% (Fig. 4f), and achieved a higher frequency of enucleation and larger size of terminal erythroid cells in each of the IVS II-654 β -thalassemia samples (Fig. 4g,h), suggesting that therapeutically correction of the pathophysiologic IVS II-654 (C>T) mutation could result in improvement of terminal erythroid maturation in vitro.

4. There are general issues with regards to data quality and formatting.

a. Some error bars appear to be too narrow given data in the manuscript. Figures 1c and 1j appear to use the same reagents and conditions, but the editing rates are much lower in 1j than 1c. If these were edited the same way, they should have been pooled and the error bars should be revised.

b. Some panels contain redundant data and should be combined. Specifically, Figure 1j and 1m appear to have the same data in the first four bar graphs, where the axis labels are changed in the latter panel.

4a. We found that variability reported as SEM instead of standard deviation caused error bars to appear too narrow. The standard deviation is a more rigorous test of statistical significance, so we re-reported the deviation using the standard deviation.

Hemoglobin synthesis in erythrocytes differentiated in vitro can be affected by cell origin and batch effects of in vitro differentiation. We controlled the conditions of each in vitro differentiation experiment to be as consistent as possible, but donor cells from different sources still affect hemoglobin synthesis in erythrocytes in the end-stage of differentiation in vitro. Therefore, to maintain the validity of the within-group comparison, we compared the experimental results of each batch of in vitro differentiation separately.

4b. Redundant data has been integrated.

Minor Concerns

1. Line 292- This is not single base substitution. There are varying probabilities that 3 sites are

changed.

We agree that not individual bases were edited and corrected accordingly in the manuscript.

2. Figure references in the text are wrong, for example:

a. Lines 113-118 – figures references are wrong, they should be f-h.

Thanks for the reminder, we have re-checked the numbering of the figures to ensure that all the numbers correspond to the correct content.

3. Figure legends are incorrect, to list a few instances:

a. Fig 4e = mean +/- s.e.m. of how many?

b. Many typos in Figure 4 legend.

c. Fig 4e legend does not define which site is which.

We have revised relevant sections of the text to remove inaccurate information.

4. There are statements in the manuscript that are inconsistent with data in the manuscript.

a. It is not interesting that the ABE8e-SpRY variant doesn't have high indel levels if the overall editing rates are lower as well.

b. Disagree with statement 239. The authors do not convince me that the SpRY construct has therapeutic potential.

c. Disagree with statement 259. The authors own data show that there is minimal if any increase in HBG and the term "globin chain balance" is nebulous and should be properly defined.

4a. We went back over the manuscript again and discovered no mention of ABE8e-SpRY indel levels. We may have some misunderstandings on this issue, please assist us in clarifying. Thanks.

4b. We found that editing efficiencies differed using sgRNAs produced by different suppliers. As mentioned earlier, the data presented in this version of the manuscript do not differentiate between providers, so that editing efficiency is generally compromised. For ABE8e&HbEsg1, the sgRNA provided by GenScript(Gen) has slightly improved editing efficiency and editing effect compared with Synthego(Syn), the editing efficiency has increased from 76.9% to 78.8%, and the proportion of β -globin in erythroid cells from in vitro differentiation has increased from 76.0% to 82.3%. For ABE8e-SpRY&HbEsg2, there was a significant improvement, the editing efficiency increased from 23.3% to 55.2%, and the proportion of β -globin in erythroid cells from in vitro differentiation increased from 20.0% to 57.5%. We also have added morphological experiments results on edited IVS2-654 cells to provide more evidence for the therapeutic potential of ABE8e-SpRY.

4c. There may be some misunderstanding here, for the IVS2-654 mutation, it was edited to repair the abnormal splicing in situ to increase the expression of normal HBB, not the level of HBG. Increasing the expression of normal HBB can improve the β -chain/ α -chain balance. The sentence containing "globin chain balance" has been removed for clarity.

5. There is a dire loss of cells in double edited secondary recipients in Extended data Fig 7b.
a. Is this due to HBG editing? Double edits? What do the authors think is happening here?

We also found this problem in our experiments, the co-electroporation of two sgRNAs is more cytotoxic than a single sgRNA, which may be the main reason for the massive cell loss. The degree of cell loss was comparable between editing HBG alone and editing the BCL11A+58 site.

Responses to Reviewer #2

Reviewer #2 (Remarks to the Author):

This manuscript by S. Chen reports progress in adapting the adenine base editor, ABE8e, for genomic alterations that induce fetal hemoglobin and correct selected β -thalassemia mutations. Together these strategies might be applied in clinically significant β -hemoglobinopathies. The first series of experiments convincingly show successful targeting of the erythroid-specific enhancer of BCL11a and HGB 1/2 promoters by optimized sgRNA complexed in an ABE8e RNP that generated high level editing and induced γ -globin expression both in vitro and in xenografted mice from human HSPCs. While neither these genomic targets for γ -globin induction by genomic editing is novel, this represents incremental progress in developing safe and effective editing reagents. While there also appears to be an additive effect by multiplex editing, this also has the potential to amplify off-target and other deleterious effects of editing. In addition, other published evidence strongly suggests that editing at a single target locus (either BCL11a or HGB 1/2) is sufficient for mitigating phenotypic expression of β -hemoglobinopathies.

The strength of the report is the progress shown in adapting ABE8e to the near PAM-less variant, ABE8e-SpRY, for which sgRNA optimization was executed to target HbE and IVS2-654

thalassemia mutations. Although, it is not yet clear if the correction rate demonstrated in the manuscript is sufficient, the possibility of using base editors in lieu of inefficient template-directed homologous repair of common β -hemoglobinopathy mutations by Cas RNPs is both significant and very interesting, and has the potential to advance the field.

Thank you for taking the time to review this manuscript and make such insightful suggestions. Detailed responses to your questions are attached below.

Comments:

1. I do not agree that BCL11A and HBG promoters can be viewed as having independent regulatory responses in the context of these experiments (Line 119 of ms). Both the targeted HBG and BCL11a genomic modifications were intended to inhibit γ -globin transcription repression by BCL11a. These are not independent actions. If the editing at both loci had independent mechanistic effects that were interactive, the multiplex editing should have generated a cooperative effect on γ -globin de-repression, which it did not. At best, the multiplex effect was additive.

Thanks to the reviewer for the reminder. We agree with the view that editing of BCL11A and HBG promoters does not have an independent effect on the elevation of γ -globin, and they both intend to suppress the repression of γ -globin transcription by BCL11a. Co-editing is additive rather than collaborative. We have made corresponding changes in the manuscript.

2. The xenotransplantation mouse strain used in these experiments (NCG-X) permitted remarkably robust human hematopoietic cell engraftment, exceeding 95% (Fig 2b). Is it possible that this level of engraftment generated anemia in the mouse at 16 weeks? If so, murine paracrine signaling could have a cross-reactive effect on human erythropoiesis in the mouse, and contribute to HbF induction in human erythroid progenitor cells. Was this possibility investigated? It might be important to repeat some of the xenografting studies in NSG mice where engraftment is not as efficient.

Thanks to the reviewer for the alternative view, we would like to make several explanations here to strengthen our manuscript. Firstly, the donor cells with a high proportion of humanization in fig2.b are derived from healthy individuals rather than thalassemia patients, and the cells of healthy donors will differentiate into large numbers of red blood cells and will not cause anemia in the transplant recipient. The proportion of humanized cells after thalassaemia patient transplantation is not particularly high, in the range of 29.1% to 82.7%. Secondly, according to our previous results (in fig3d. h)¹, in sickle cells, regardless of whether the proportion of humanization is high or low, the expression level of γ in unedited cells is very low, while the expression level of γ in edited cells is significantly increased. This indicates that the proportion of human cells does not affect the induction of γ . Finally, according to our data in fig3d.f., in our NCG-X mouse xenograft system, thalassaemia patient cells were transplanted with a moderate proportion of humanization. And the editing group still showed a higher γ induction compared with the mock group under the condition of relatively low humanization ratio. It is shown that gene editing, rather than a high proportion of humanization, is the main

reason for γ -globin induction.

3. The assessments of γ -globin expression (as a proportion of total β -like globin expression) in Fig 2d and Ext data Fig 7 should use the same scale to facilitate comparison of expression in primary and secondary xenografts. The secondary mice appear to show a smaller contribution in γ -globin expression, which could indicate a problem with HSC propagation.

Individual differences in mouse erythroid differentiation, variations in the humanization ratio, and differential amplification of sequential transplantation result in widely varying in γ -globin induction in secondary transplantation. So different scales were adopted to facilitate demonstration.

The low γ -globin induction in the second transplant was an anticipated outcome given that the first transplantation used edited HSPCs, and the second transplantation used total bone marrow cells from the first transplantation, of which only a small fraction of cells were HSPCs. The primary goal of the secondary transplantation is to show that the long-term hematopoietic stem cells can be successfully edited with our base editing system.

Additionally, double editing will reduce the humanization ratio in secondary transplantation due to cytotoxicity, which in turn affects the maturity of erythrocyte differentiation. This results in a decrease in the induction of gamma-globin expression in secondary mice. Subsequent studies can optimize the sgRNA purification process and the delivery of editing reagents to reduce toxicity while maintaining high γ -globin induction, thereby advancing clinical applications.

4. Based on the data in Fig 3b (with HSPCs from a $\beta^0\beta^+$ thalassemia donor) and presented in Line 205, it is important to note that improving the β -globin to α -globin ratio from 0.5 to 0.7 is not sufficient to overcome all the deleterious effects of globin chain imbalance and ineffective erythropoiesis.

We agree with the reviewers that increasing the ratio of β -globin to α -globin from 0.5 to 0.7 is not sufficient to overcome all the deleterious effects of globin chain imbalance and ineffective erythropoiesis. We have revised the corresponding description in the manuscript to indicate "alleviate" rather than "complete elimination" of thalassaemia symptoms. In addition, it is still uncertain whether the balance of human globin chains can be accurately reflected in the immunodeficient mouse xenograft model, and the results of in vitro differentiation (fig. 3b) may be more informative.

5. The HSPCs from the E/ β thal donor appeared to have no β -like globin expressed in unedited cells (Fig 4b). HbE/ β thal disease is very heterogenous, according in part to whether there is β^0 or β^+ allele. The genotype of the HbE donor was not reported, and ideally HSPCs from several patients should be analyzed. The same would be true for HSPC donors with the IVS2-654 mutation.

We have indicated the donor cells used in the figure legend summarized the genotypes of various donor cells and added them to the SI data.

6. The Fig 4 heading (... therapeutic editing in β -thalassemia/HbE patient...) also presents data IVS2-654 splice junction editing. The heading should be corrected to include both genotypes.

Thanks for the reminder, the heading has been corrected.

7. The paragraph in the discussion that begins at line 308 reports primary off-target data, which should be moved to the results section and referred to in the discussion.

Thanks to the reviewer's suggestion, we have revised the manuscript accordingly.

8. Some of the xenotransplantation studies used just 3 mice, others 5, but in every group the variability was reported as SEM in lieu of standard deviation. The standard deviation more rigorously tests for statistical significance and should be used to assign significance in comparisons.

We agree with the reviewer's proposal and we have made emendations in the manuscript.

1. Wu, Y, *et al.* Highly efficient therapeutic gene editing of human hematopoietic stem cells. *Nat Med* **25**, 776-783 (2019).

REVIEWERS' COMMENTS

Reviewer #1 (Remarks to the Author):

The authors have addressed my comments and I find the manuscript to be much improved. I look forward to seeing this in press.

Reviewer #2 (Remarks to the Author):

This revised, improved manuscript is satisfactory and is responsive to the initial critiques. It is very interesting to note the effect of sgRNA source on editing efficiency, which is an important observation for those working in this field.